# Genome-Wide Identification of DUF668 Gene Family and Expression Analysis under Drought and Salt Stresses in Sweet Potato [*Ipomoea batatas* (L.) Lam]

**DOI:** 10.3390/genes14010217

**Published:** 2023-01-14

**Authors:** Enliang Liu, Zhiqiang Li, Zhengqian Luo, Linli Xu, Ping Jin, Shun Ji, Guohui Zhou, Zhenyang Wang, Zhilin Zhou, Hua Zhang

**Affiliations:** 1Grain Crops Institute, Xinjiang Academy of Agricultural Sciences, Urumqi 830000, China; 2Xinjiang Academy of Agricultural Sciences, Comprehensive Proving Ground, Urumqi 830052, China; 3Adsen Biotechnology Co., Ltd., Urumqi 830000, China; 4College of Agriculture, Tarim University, Tarim Rd, Alaer 843301, China; 5Xuzhou Institute of Agricultural Sciences in Xuhuai District, Xuzhou 221131, China; 6School of Life Sciences, Xinjiang Agricultural University, Urumqi 830022, China

**Keywords:** sweet potato, DUF668 gene family, bioinformatics, qRT–PCR

## Abstract

The domain of unknown function 668 (DUF668) is a gene family that plays a vital role in responses to adversity coercion stresses in plant. However, the function of the DUF668 gene family is not fully understood in sweet potato. In this study, bioinformatics methods were used to analyze the number, physicochemical properties, evolution, structure, and promoter *cis*-acting elements of the IbDUF668 family genes, and RNA-seq and qRT–PCR were performed to detect gene expression and their regulation under hormonal and abiotic stress. A total of 14 IbDUF668 proteins were identified in sweet potato, distributed on nine chromosomes. By phylogenetic analysis, IbDUF668 proteins can be divided into two subfamilies. Transcriptome expression profiling revealed that many genes from DUF668 in sweet potato showed specificity and differential expression under cold, heat, drought, salt and hormones (ABA, GA3 and IAA). Four genes (*IbDUF668-6, 7, 11* and *13*) of sweet potato were significantly upregulated by qRT-PCR under ABA, drought and NaCl stress. Results suggest that the DUF668 gene family is involved in drought and salt tolerance in sweet potato, and it will further provide the basic information of DUF668 gene mechanisms in plants.

## 1. Introduction

Sweet potato is an important fodder and industrial raw material crop, most of which is planted in semiarid areas because of its strong adaptability and resistance to drought and barrenness [1]. In recent years, with the increase in global climate warming and soil salinization, sweet potato has become susceptible to drought and salt stress during key growth periods, such as the slow-rooting and root-expansion periods, and has become the main limiting factor for the improvement of sweet potato yield [2,3,4]. Studies have shown that salt and drought stress inhibit the growth of sweet potato leaves, vines and roots, and decrease in sweet potato tuber yield, with the greatest reduction being from stress at the root branching stage [5,6]. Many studies have been performed on the effect of different degrees of salt and drought stress on the growth of sweet potato, and it is believed that severe drought and salt stress increase soil mechanical resistance, affect the absorption of nutrients by the root system, limit the expansion of tuber roots, and lead to decreased sweet potato root growth and dry matter accumulation [4,5,6]. Due to the rapid development of genome sequencing, the genome of sweet potato has made gratifying progress, which helps understand the stress adaptation regulation in sweet potato [7].

In recent years, as the research has deepened, the number of domains of unknown function (DUF) superfamily members has increased rapidly [8,9]. *AGM1*, *AGM2* members of the DUF579 family are required for glucuronic acid 4-O-methylation of highly glycosylated arabinogalactan proteins (AGPs) in *Arabidopsis thaliana* [10]. *AT3g55990* from the DUF231 gene family is a novel negative regulator of cold acclimation [11]. Overexpression of the *OsDUF946.4* and *OsSIDP366* gene in rice enhances tolerance to high salt and drought [12,13]. Overexpression of *TaSRHP*, a gene of the DUF581 from wheat, can improve salt tolerance in *Arabidopsis thaliana* [14].

Although some genes from the DUF gene family have been identified, a large number of DUF668 members are still unknown. Besides, research on the DUF668 gene family is currently mainly focused on model plants, while comprehensive analysis of the DUF668 gene family is rarely reported [15,16,17]. However, studies on this gene family are limited and have only been identified in *Arabidopsis thaliana*, rice and cotton; however, the function, taxonomy and evolution of this gene family have not been systematically studied in sweet potato [8,9]. In our study, DUF668 family genes were identified based on the sweet potato genomic data system, and bioinformatic analysis was performed. The chromosomal distribution, evolution, *cis*-acting elements of promoter, and expression profiles of the DUF668 gene under different stresses and hormone treatments were analyzed. qRT–PCR was performed to detect the expression of candidate genes under drought and salt stress and abscisic acid (ABA) induction and revealed some of their unknown biological functions. The results lay a foundation for further analysis of their functions.

## 2. Materials and Methods

### 2.1. Plant Material

Sweet potato (Jishu 25) was used in this study. It was grown in the crop research greenhouse of the Xinjiang Academy of Agricultural Sciences from October to December 2021. The seedlings of Jishu 25 were grown in Hoagland solution at 25 °C with 16 h light/8 h dark treatment. Seedlings at the 5–6 leaves stage were treated with 15% PEG and 20% NaCl, and the leaves were sprayed with 100 µm/L ABA. Root tissues at 8–10 cm length were treated with 15% PEG and 20% NaCl. Leaf tissues after being treated with ABA were collected at 0, 3, 6, 12 and 24 h, respectively. Three biological replicates for each sample were frozen by liquid nitrogen and stored at −80 °C for further analysis.

### 2.2. Identification and Bioinformatics Analysis of the IbDUF668 Gene

The genomic and proteomic data of sweet potato were gained from the database of sweet potato (http://sweetpotato.plantbiology.msu.edu/, accessed on 21 December 2021). The *Arabidopsis thaliana* genome and protein sequences were downloaded from the *Arabidopsis thaliana* Genome Database TAIR website (https://www.arabidopsis.org/, accessed on 21 December 2021). The genome of rice was downloaded from the Rice Genome Database (http://rice.plantbiology.msu.edu/, accessed on 21 December 2021). The protein sequences in sweet potato were identified by the hidden Markov model (PF05003) of the DUF668 gene domain and the hidden Markov model (HMM) in HMMER3.0 software (http://hmmer.org/, accessed on 21 December 2021) [18]. After removing the redundancy, all candidate genes were verified in the NCBI (https://www.ncbi.nlm.nih.gov/cdd/, accessed on 22 December 2021) database. ExPASy software (http://cn.expasy.org/tools, accessed on 22 December 2021) was used to calculate the number of amino acid residues, relative molecular mass, and the theoretical isoelectric point of the IbDUF668 protein. EuLoc software (http://euloc.mbc.nctu.edu.tw/, accessed on 22 December 2021) was used to predict the subcellular localization of the IbDUF668 protein [19,20].

### 2.3. Collinear Analysis and Phylogenetic of the IbDUF668 Gene Family

DUF668 protein sequences in *Arabidopsis thaliana*, rice, cotton and sweet potato were aligned using MEGA 9 software. Then, based on the alignment results, a phylogenetic tree was constructed using the proximity method (bootstrap value = 1000) [21]. The results were beautified with the online tool Evolview (https://evolgenius.info/, accessed on 23 December 2021) [22].

### 2.4. Chromosomal Location, Gene Structure and Motif Analysis of the IbDUF668 Gene Family

The chromosomal location information of IbDUF668 family genes was obtained by TBtools software. Map of the chromosomal location of the DUF668 gene was drawn by Mapchart software. The phylogenetic tree of IbDUF668 was constructed by MEGA 9 software. Motif analysis was used through the MEME program (functional domains = 10). Above results were visualized by using TBtools software [23].

### 2.5. Analysis of Upstream Cis-Acting Elements of the IbDUF668 Gene

A 2000 bp DNA sequence upstream of the IbDUF668 gene was gained by TBtools software. Possible *cis*-acting elements were predicted using the PlantCARE database (http://bioinformatics.psb.ugent.be/webtools/plantcare/html/, accessed on 23 December 2021). R language and TBtools software were used to visualized the possible *cis*-acting elements [23,24].

### 2.6. RNA-Seq Analysis

Whole sweet potato in vitro plants, drought-treated transplant to mannitol medium, 14 h photoperiod 28 °C day/22 °C night, cold-treated grown at 14 h photoperiod 10 °C day/4 °C night, heat-treated at 14 h photoperiod 35 °C day/35 °C night, salt-treated transplant to NaCl medium, 14 h photoperiod 28 °C day/22 °C night, (shoots, petioles, and leaves) pooled at 24 h. Whole in vitro plants, transplant to ABA, GA3 and IAA medium, 14 h photoperiod 28 °C day/22 °C night, (shoots, petioles, and leaves) pooled at 24 h. Controls were transplanted to standard shooting medium, 14 h photoperiod 28 °C day/22 °C night, pooled at 24 h. All samples were from the same batch of plant material. Sweet potato tissue (root, stem, leaf, flower and mosaic), hormones (ABA, gibberellin (GA) and auxin (IAA)) and adversity (cold, heat, drought) from NCBI (https://www.ncbi.nlm.nih.gov/, accessed on 21 December 2021) database and salt gene expression profiling data of DUF668 after stress. The expression heatmap was drawn using the R language pheatmap package.

### 2.7. Total RNA Extraction, cDNA Synthesis, and qPCR

Primers of the IbDUF668 gene were designed in the region of the 5′ or 3′ end of the gene sequence using Primer 6.0 software (Appendix A). Root and leaf tissue cDNA were used as templates, expression of candidate genes was measured by qRT–PCR, and each sample had three replicates. Reverse transcription was performed using an M-MLV RTase cDNA Synthesis Kit (TaKaRa, Japan). qRT–PCR was performed using the Roche LightCycler^®^ 480II System under the following conditions: 95 °C 15 s, followed by 40 cycles of 95 °C 15 s, 55 °C 15 s, and 72 °C 15 s. Relative quantification was performed using the 2^–ΔΔCt^ method [25].

## 3. Result

### 3.1. Identification of DUF668 Gene Family in Sweet Potato

The DUF668 gene was first comprehensively searched for in the sweet potato genome using HMMsearch. The search results were validated against the NCBI database (Appendix A). Fourteen DUF668 protein sequences were contained in sweet potato. We named *IbDUF668-1*~*IbDUF668-14* according to the chromosomal positions of the 14 sequences of sweet potato. The length of the IbDUF668 family gene open reading frame (ORF) is 1389–1947 bp, and 462–648 amino acid residues were contained in the encoded protein. Relative molecular mass is 51.79–71.84 kDa, and theoretical isoelectric point is 4.60–10.21. Subcellular localization of the proteins showed 10 in the nucleus, two in the chloroplast, one in the endomembrane and one in the organelle membrane (Table 1).

Fourteen IbDUF668 genes were distributed on nine chromosomes (Chr01, Chr03, Chr04, Chr05, Chr06, Chr08, Chr09, Chr13 and Chr15) of sweet potato (Figure 1), of which Chr04 and Chr13 staining each contained three DUF668 genes, and Chr15 contained two DUF668 genes; the rest of the chromosomes contain only one DUF668 gene. Although Chr01 has the longest length, it contains only one DUF668 gene. These findings suggest that there is no direct relationship between IbDUF668 gene distribution and chromosome length.

### 3.2. Evolutionary Analysis of the Sweet Potato DUF668 Gene

To better understand the relationship of evolution on IbDUF668, a phylogenetic tree of the 14 IbDUF668 proteins was constructed: six *Arabidopsis thaliana* DUF668 proteins, 12 rice DUF668 proteins and 32 cotton DUF668 protein sequences (Figure 2A). The phylogenetic tree was divided into two groups according to the grouping of *Arabidopsis thaliana* and rice. The Group 1 and Group 2 subgroups contained nine and five IbDUF668 proteins, respectively. Among them, the number of IbDUF668 proteins in the Group 1 subgroup was six more than that in *Arabidopsis thaliana*, three more than that in rice, and 13 less than that in cotton. The number of IbDUF668 proteins in the Group 2 subgroup was two more than that in *Arabidopsis thaliana*, one less than that in rice, and five less than that in cotton. This indicates that during the evolution of the IbDUF668 gene family, the genes of Group 1 have been significantly expanded, while the genes of Group 2 have been significantly lost. This may have much to do with the planting environment of the crop and the part where the fruit matures. Although there are relatively few members in Group 2, they have been retained during the evolution of sweet potato, revealing that they may play an important role in processes of biological activity. In the evolutionary tree, sweet potato and cotton proteins are the closest, suggesting that DUF668 may function more similarly between the two species.

To explore the evolutionary relationship of the IbDUF668 gene family, we constructed a collinear relationship of sweet potato with DUF668 genes in *Arabidopsis thaliana*, rice and cotton (Figure 2B). We found that each DUF668 family gene from sweet potato formed five pairs of colinear genes with three genes from *Arabidopsis thaliana* and seven pairs of colinear genes with four genes from rice and showed the best collinearity with cotton. Consistent with the results of the phylogenetic tree, although the number of DUF668 genes in *Arabidopsis thaliana* and rice is significantly less than that in sweet potato, two or more IbDUF668 genes are used for the same DUF668 in *Arabidopsis thaliana* and rice. The number of DUF668 genes in *Arabidopsis thaliana* and rice is small, but each gene may contain more biological functions, which in turn regulate the growth and development of *Arabidopsis thaliana* and rice under stress, which also shows the complex function of DUF668 family genes.

### 3.3. Evolutionary Tree, Gene Structure and Motif Analysis of Sweet Potato DUF668 Gene

To further understand the composition of the IbDUF668 gene, we compared its gene structure. The Group 1 subgroups all have only one exon, while the Group 2 subgroups all contain 12 exons (Figure 3). All IbDUF668 proteins contain the DUF668 domain (motif 1). Among them, all DUF668 proteins contain a DUF3475 domain at the 5′ end, which indicates that the two gene families DUF668 and DUF3475 may contain the same gene (Appendix A), which is consistent with the research results of other species, such as *Arabidopsis thaliana* and rice. All members of IbDUF668 are divided into two subgroups according to the evolutionary tree. DUF668 gene in Group 1 has two fewer motif structures (motif 4 and motif 9) than the genes in the Group 2 subgroup, and they are distributed in the middle of the genes. However, motif 10 is unique to Group 1 and is mainly distributed at the 3′ end. The DUF668 gene of the Group 2 subgroup is longer than that of Group 1 and contains more motifs and exons, which indicates that the distribution of motifs in sweet potato DUF668 is highly correlated with gene length.

### 3.4. Analysis of Promoter Cis-Acting Elements of DUF668 Gene in Sweet Potato

Transcription factors (TFs) refer to proteins capable of binding DNA in a sequence-specific manner and regulating transcription, which in turn regulates plant functions, including responses to environmental factors and growth and development [26,27]. Analysis of promoter *cis*-elements of the IbDUF668 gene revealed that a variable number of *cis*-acting elements were found in response to phytohormones and environmental stress (Figure 4 and Appendix A). Different types and numbers of cis-acting elements were contained in different IbDUF668 genes; this finding indicates that different IbDUF668 families may exert their biological functions through different signaling pathways.

### 3.5. RNA-Seq Analysis of the Sweet Potato DUF668 Gene

We analyzed tissue-specific expression analysis and hormone-induced and stress-induced expression patterns of 14 IbDUF668 genes (Figure 5). Tissue-specific analysis results (Figure 5A) showed different DUF668 genes were specifically expressed in different tissues. The root is the main organ for plants to absorb water and mineral elements from soil, which marks the evolution and development of plants from aquatic to terrestrial. Among them, five genes (*IbDUF668-4, 7, 6* and *8*) were highly expressed in roots, indicating that they may play a vital role in the process of absorption of nutrients and stress in sweet potato.

In this study, the expression profiles after ABA, GA3 and IAA treatments were analyzed (Figure 5B). Results suggested that the expression patterns of IbDUF668 genes changed significantly under different hormone treatment conditions, but most of the genes were induced by the hormone ABA, and the induction patterns of different genes were completely different. Seven IbDUF668 genes (*IbDUF668-1, 6, 7, 9, 10, 11, 12*, and *13*) were significantly more affected by ABA than other hormones, and the remaining six were highly expressed except *IbDUF668-10*. Three IbDUF668 genes (*IbDUF668-4, 5* and *8*) were significantly more affected by GA3 than other hormones. Although IAA can also induce changes in DUF668 gene expression, only IbDUF668-2 expression changes under the induction of IAA are significantly higher than those of other hormones.

Expression analysis of the IbDUF668 gene under heat, cold, drought and salt treatment (shown in Figure 5C), among which drought, salt and cold stress can induce more DUF668 gene expression, and among which eight IbDUF668 genes (*IbDUF668-1, 3, 4, 5, 8, 9, 10* and *14*) showed significant changes in the expression levels after cold stress but basically did not change before and after heat, drought and salt stress. Compared with other abiotic stresses, the expression of five IbDUF668 genes (*IbDUF668-2, 6, 5, 11* and *13*) changed significantly under drought stress. Compared with drought and cold stress treatments, the low IbDUF668 gene expression induced by heat and salt stress indicates that the IbDUF668 gene may be more important for drought and cold stress. Expressions of *IbDUF668-7*, *IbDUF668-12* and *IbDUF668-14* were significantly changed under salt stress, suggesting that different IbDUF668 genes may have different functions in the resistance of abiotic stress in sweet potato.

### 3.6. QRT–PCR of the Sweet Potato DUF668 Gene under Exogenous ABA, Drought and NaCl Stress

With changes in climate and the soil environment, drought and salt stress have become the most important abiotic stresses faced by sweet potato, and ABA is believed to be involved in this process [2,3,4,5]. We selected 10 IbDUF668 (*IbDUF668-2, 3, 6, 7, 9, 10, 11, 12, 13* and *14*) genes performed qRT–PCR (Figure 6). Under ABA treatment, the six genes (*IbDUF668-2, 3, 6, 7, 11*, and *13*) were significantly induced at different periods compared with 0 h, indicating that these genes may be involved in sweet potato response to drought stress. Previous studies have shown that exogenous ABA can improve the resistance of plants to abiotic stress, and abiotic stress can also induce an increase in the hormone ABA in plants. These results show that *IbDUF668-2, IbDUF668-3, IbDUF668-6, IbDUF668-7, IbDUF668-11*, and *IbDUF668-13* may play an important role in the anti-abiotic stress process of sweet potato through the ABA pathway.

To investigate the role of the IbDUF668 gene in drought, we analyzed the expression patterns of these 10 genes under PEG stress (Figure 7). Under drought treatment, compared with 0 h, the RNA transcription levels of five genes (*IbDUF668-6, 7, 10, 11* and *13*) were significantly induced at different periods, and the expression levels were significantly increased, indicating that these genes may be involved in the process of drought stress in sweet potato. These five genes were significantly upregulated both under drought stress and exogenous ABA treatment, with a maximum value at 12 h on the large island. Results suggest that these five genes may play a role in the response to drought stress in sweet potato.

To investigate the role of the IbDUF668 gene in salt stress, we analyzed the expression patterns of these 10 genes under NaCl stress (Figure 8). Under salt stress, compared with 0 h, the RNA transcription levels of eight genes (*IbDUF668-2, 6, 7, 9, 11, 12, 13* and *14*) were significantly induced in different periods, and the expression levels were all significantly increased, indicating that these genes may be involved in the process by which sweet potato responds to salt stress. Among them, four genes (*IbDUF668-6, 7, 11* and *13*) were significantly upregulated under drought and salt stress or under exogenous ABA treatment and reached the maximum value at 12 h. These results suggest that these four genes may play a role in the response of sweet potato to drought and salt stress through an ABA-dependent pathway.

## 4. Discussion

The DUF668 gene plays an important role in plant growth and development and stress [8,9]. With the continuous improvement of various plant genome works, the DUF668 gene family in cotton, *Arabidopsis thaliana* and rice has been studied, and it was found that the number of family members is related to the size of the genome [8,9]. The polyploidy event or the whole genome duplication event directly doubles the chromosome, which is considered to be the driving force of species differentiation. During the process of this differentiation, the gene family produces multiple members of the gene family [28,29,30]. Moreover, we did not find a tandem duplication events phenomenon in sweet potato, suggesting that the DUF668 gene family may be conserved in the evolution of plants. Fourteen members of the IbDUF668 gene family were identified, which is higher than those in rice and *Arabidopsis thaliana*, and fewer than those in cotton, possibly due to the size of the genome. IbDUF668 was distributed on nine chromosomes of sweet potato, and there was no obvious tandem duplication phenomenon, which indicated that tandem duplication occurred less frequently in the expansion and evolution of the DUF668 gene family.

We found that although intronless genes are a typical feature of prokaryotic genes, they also occupy a certain proportion in eukaryotes, such as 21.7% in *Arabidopsis thaliana* and 19.9% in rice. A large number of intronless genes originate from prokaryotes and are replicated in plant genomes [31,32,33]. Analysis of the gene structure of the IbDUF668 family found that eight IbDUF668 genes had no introns and only six IbDUF668 genes had multiple introns (Figure 3). The intronless DUF668 gene also occupies a higher proportion in other species, such as cotton, *Arabidopsis thaliana* and rice [8,9]. It reveals that the structure of DUF668 family genes is highly conserved in different species. The specific domains and protein motifs of the DUF668 protein determine that it can recognize and bind to different DNAs, thereby participating in different regulatory pathways and performing different biological functions (Figure 3). In this study, most of the motifs contained in DUF668 proteins of different subfamilies were the same, and some protein motifs showed specific distributions in different subfamilies. The arrangement sequence is similar, and there are differences in the arrangement in different subfamilies, leading to speculation that genes in the same subfamily may have similar functions [34,35]. Although there are fewer genes in Group 2, the DUF668 gene in the Group 2 subgroup is longer than that in Group 1 and contains more motifs and exon structures, indicating that IbDUF668 in Group 2 may have more complex functions.

The DUF668 protein has been found in cotton, *Arabidopsis thaliana* and rice and is widely involved in the regulation of a variety of stress responses with diverse functions [8,9]. Currently, the functions of the DUF668 gene family in model plants have not been verified [8,9]. In this study, the promoter *cis*-acting elements of the IbDUF668 gene were analyzed, and it was found that hormone-responsive elements and stress-responsive elements widely exist in the promoter (Figure 4). Among them, an abscisic acid-binding site has the most *cis*-acting elements, and ABA has been widely studied in plant stress resistance [36,37]. This implies that he IbDUF668 may have a regulatory effect that depends on the ABA pathway. 

From the transcriptome sequencing data, it can be seen that the IbDUF668 family genes can respond to salt and drought stresses to different degrees, and the expression patterns of some genes distributed in the same subfamily are similar. The results of the transcriptome expression profiling in this study showed that the IbDUF668 gene could respond to drought and salt stress and be induced by ABA (Figure 5). The increase in the amount of plant root exudates, especially organic acids, under drought and salt stress may contribute to plant resistance. Salt and drought stresses will significantly change their plant root structure and composition to ensure normal plant growth [38,39,40,41]. The responses of different DUF668 genes to different stresses were different, indicating that the IbDUF668 gene could be involved in the regulation of various adversity stresses (Figure 5). It is worth noting that four differentially expressed genes (*IbDUF668-6, 7, 11* and *13*) were screened according to gene expression under ABA, salt and drought stress, and these four genes were expressed under drought and salt stress or exogenous stress. Their expression was significantly upregulated under ABA treatment (Figure 6, Figure 7 and Figure 8). Previous studies have shown that exogenous ABA can improve the salt and drought tolerance of plants, and salt and drought stress can also induce an increase in the ABA content in plants [42,43]. These four genes contain a variable number of ABA *cis*-acting elements (Figure 4). These comprehensive results suggest that these four genes may play an important role in sweet potato drought resistance through the ABA pathway.

## 5. Conclusions

A genome-wide identification of the IbDUF668 gene was performed. The IbDUF668 gene family was found to be conserved in the evolution of sweet potato based on gene number, chromosomal location and evolutionary analysis. Promoter *cis*-acting element analysis revealed that the DUF668 gene plays an important role in regulating the growth and development of sweet potato and in responding to stress. Expression analysis showed that *IbDUF668-6, IbDUF668-7, IbDUF668-11* and *IbDUF668-13* play important roles in the sweet potato response to drought and salt stress. 

## Figures and Tables

**Figure 1 genes-14-00217-f001:**
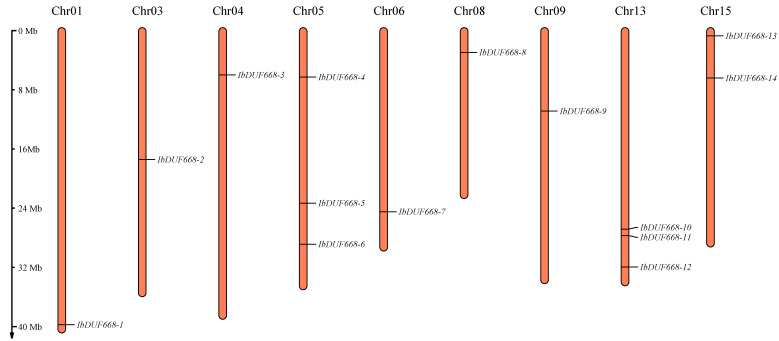
Chromosome location of sweet potato DUF668 genes.

**Figure 2 genes-14-00217-f002:**
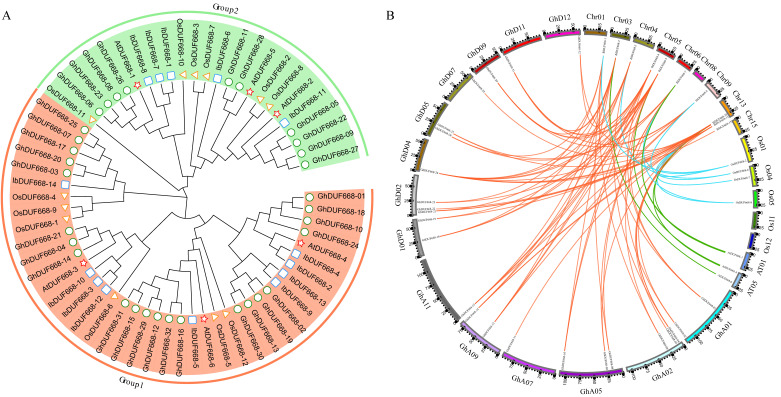
Evolutionary analysis of the sweet potato DUF668 gene. (**A**) shows the phylogenetic tree in *Arabidopsis thaliana*, rice, cotton and sweet potato, (**B**) DUF668 gene collinearity in *Arabidopsis thaliana*, rice, cotton, sweet potato.

**Figure 3 genes-14-00217-f003:**
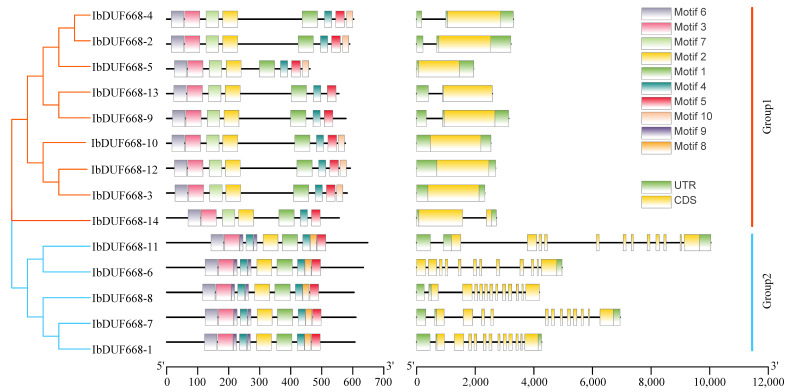
Evolutionary tree, gene structure and conservative motif analysis of the DUF668 gene family in sweet potato.

**Figure 4 genes-14-00217-f004:**
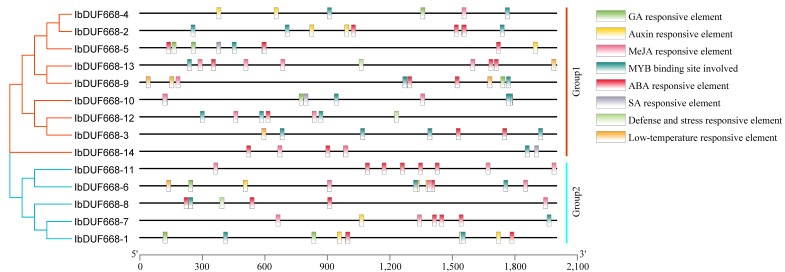
Analysis of promoter *cis*-acting elements of the IbDUF668 family genes.

**Figure 5 genes-14-00217-f005:**
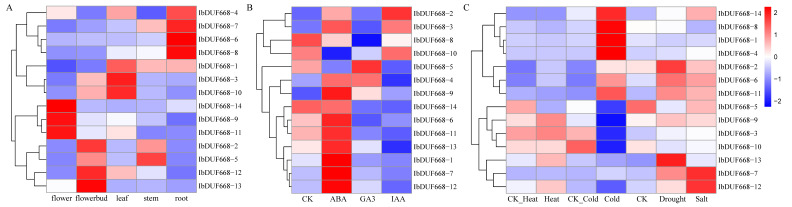
Expression analysis of DUF668 genes in sweet potato. (**A**) Tissue-specific expression analysis, (**B**) ABA, GA3 and IAA-induced expression analysis, (**C**) Expression analysis under hot and cold drought and salt stress.

**Figure 6 genes-14-00217-f006:**
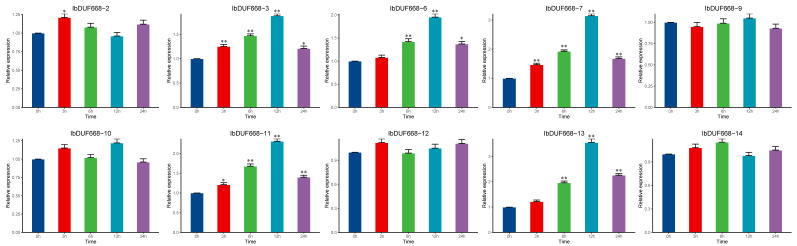
Tissue-specific expression analysis of the DUF668 gene in sweet potato under ABA. Error bars represent the average of three replicates ± SE (* *p* < 0.05; ** *p* < 0.01).

**Figure 7 genes-14-00217-f007:**
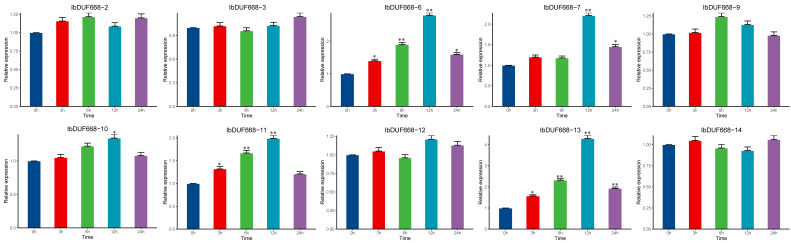
Expression analysis of the DUF668 gene in sweet potato under drought. Error bars represent the average of three replicates ± SE (* *p* < 0.05; ** *p* < 0.01).

**Figure 8 genes-14-00217-f008:**
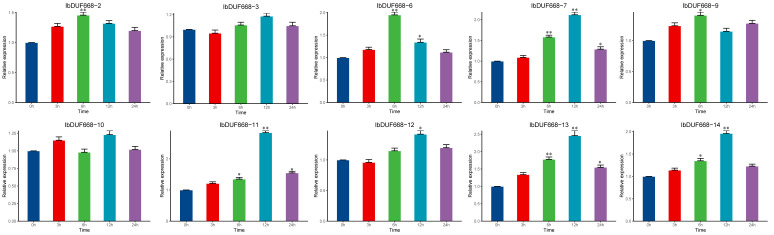
Expression analysis of the DUF668 gene in sweet potato under NaCl. Error bars represent the average of three replicates ± SE (* *p* < 0.05; ** *p* < 0.01).

**Table 1 genes-14-00217-t001:** Information of the DUF668 gene family members in sweet potato.

Gene Name	Gene ID	Open Reading Frame/bp	Protein Length/aa	Relative Molecular Weight (r)/kDa	Theoretical Isoelectric Point (pI)	Subcellular Localization
itb01g35230	IbDUF668-1	1824	607	66.99	9.39	nucleus
itb03g18020	IbDUF668-2	1773	590	66.06	9.53	nucleus
itb04g07990	IbDUF668-3	1749	582	65.52	8.16	nucleus
itb05g05910	IbDUF668-4	1812	603	67.50	9.49	nucleus
itb05g14710	IbDUF668-5	1389	462	51.79	10.32	nucleus
itb05g21500	IbDUF668-6	1905	634	71.40	6.73	nucleus
itb06g19930	IbDUF668-7	1833	610	68.31	8.89	nucleus
itb08g03240	IbDUF668-8	1815	604	66.99	9.51	nucleus
itb09g14900	IbDUF668-9	1737	578	64.62	8.99	endomembrane
itb13g18340	IbDUF668-10	1731	576	65.24	7.92	nucleus
itb13g19130	IbDUF668-11	1947	648	71.84	9.93	nucleus
itb13g24470	IbDUF668-12	1779	592	66.68	7.67	chloroplast
itb15g01020	IbDUF668-13	1668	555	62.17	9.94	organelle membrane
itb15g08500	IbDUF668-14	1671	556	63.17	9.58	chloroplast

## Data Availability

All related data in this study from NCBI (https://www.ncbi.nlm.nih.gov/ (accessed on 23 December 2021)) database and the accession number is SRP162021 (https://www.ncbi.nlm.nih.gov/sra/?term=SRP162021 (accessed on 23 December 2021)).

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
