# Peer review of "Genome-Wide Identification of DUF668 Gene Family and Expression Analysis under Drought and Salt Stresses in Sweet Potato [Ipomoea batatas (L.) Lam]"

_genes, 2023, doi:10.3390/genes14010217_

Round 1

Reviewer 1 Report

The present work exploited bioinformatics methods to analyze the characteristics of the DUF668 gene in sweet potato. The results obtained allowed the authors to suggest that this gene could have a major role in the responses of this plant to water and salt stress and therefore in its development. Overall, the document is quite well written and documented. It reflects a certain consistency in the presentation of the results obtained. It places greater value on bioinformatics, which is a booming field of scientific activity. It reflects that bioinformatics resources could have a tangible role in the development of research in several fields, including biology and medicine.

Nevertheless, here are some comments:

·         Regret not being able to have more explanation on the mechanisms of action of the gene and much more illustration. ·         Degree of reproducibility of in vitro experiments and of the results obtained!!

·         Lack of prospects !!

Author Response

Dear reviewer 1,

Thank you very much for your comments to our manuscript! According to your comment, We have revised our manuscript and identified it in red and have updated it in this revision.. The responses we provided were marked with red color.

The present work exploited bioinformatics methods to analyze the characteristics of the DUF668 gene in sweet potato. The results obtained allowed the authors to suggest that this gene could have a major role in the responses of this plant to water and salt stress and therefore in its development. Overall, the document is quite well written and documented. It reflects a certain consistency in the presentation of the results obtained. It places greater value on bioinformatics, which is a booming field of scientific activity. It reflects that bioinformatics resources could have a tangible role in the development of research in several fields, including biology and medicine.

Nevertheless, here are some comments:

Regret not being able to have more explanation on the mechanisms of action of the gene and much more illustration. Degree of reproducibility of in vitro experiments and of the results obtained!

Thank you very much for your suggestion. Research on the DUFF668 gene family is limited and only reported on model plants and cotton (doi:10.1186/s12864-021-07716-w, doi: 10.3390/genes10120980.). In rice, studies have found that the OsDUF668 genes are ubiquitously expressed in analyzed rice tissues, and seven genes show tissue-specific high expression profiles. All OsDUF668s respond to drought, and some genes resist salt, wounding, and rice blast with rapidly altered expression patterns. Some important crops have not yet been reported, and I believe that our report will provide some reference value for the study of the DUFF668 gene family. We are also designing overexpression and gene knockout experiments to complete the functional verification of IbDUF668-6, IbDUF668-7, IbDUF668-11 and IbDUF668-13 with the help of reverse genetics, and these results will be presented in follow-up studies.

Reviewer 2 Report

The manuscript named “Genome-wide identification of the sweet potato [Ipomoea batatas (L.) Lam] DUF668 gene family and expression analysis under drought and salt stress” is not well-convincing. Significant data are missing (Figures, supplementary data). In this case, I cannot review the results and discussion parts.

On the bright side, the introduction goes smoothly by pointing out the problem with recent references. The objectives and the novelty are strongly present. The material and methods are standard for this kind of study.

In my opinion, the authors should submit all the data for a revision (phylogenetic tree, chromosomal location, gene structure, motif analysis, analysis of cis-acting elements, expression analysis, sequence characteristic, the expression profile of the genes, qRT-PCR result).

Author Response

Dear reviewer2,

Thank you very much for your comments to our manuscript! According to your comment, We have revised our manuscript and identified it in red and have updated it in this revision.. The responses we provided were marked with red color.

The manuscript named “Genome-wide identification of the sweet potato [Ipomoea batatas (L.) Lam] DUF668 gene family and expression analysis under drought and salt stress” is not well-convincing. Significant data are missing (Figures, supplementary data). In this case, I cannot review the results and discussion parts.

On the bright side, the introduction goes smoothly by pointing out the problem with recent references. The objectives and the novelty are strongly present. The material and methods are standard for this kind of study.

In my opinion, the authors should submit all the data for a revision (phylogenetic tree, chromosomal location, gene structure, motif analysis, analysis of cis-acting elements, expression analysis, sequence characteristic, the expression profile of the genes, qRT-PCR result).

Thank you very much for the suggestion. This is indeed our mistake. We have committed significant data (Figures, supplementary data) and manuscript separately for the first time. We have supplemented all results in the manuscript and updated them in this upload.

Round 2

Reviewer 2 Report

I’m pleased to continue review (round 2) the manuscript entitled “Genome-wide identification of the sweet potato [Ipomoea batatas (L.) Lam] DUF668 gene family and expression analysis under drought and salt stress” and recommend publishing it as a journal article.

Foremost the authors have improved the manuscript, in comparison with the last version, adding the necessary data as tables and figures, and in my view, remain some minor revisions that I would recommend handling before publication.

Lines 83-86: This paragraph should be in the conclusion and perspectives section.

Line 130: “cis-acting” instead of “cis-acting”.

Lines 137-142: Very complex paragraph that need to be rewritten.

Line 174: “Chr05” not “Chr04”.

Line 175: “Chr15 contained 2 DUF668 genes” not 3.

Line 216: “The majority of group 1 genes have 1 exon” not all of them.

Figure 2B: Not clear and should be improved.

Author Response

Dear reviewer 2,

Thank you very much for the new round of careful review of our manuscript, for your very good comments! According to your comment, we have revised our manuscript and have updated it in this revision. The responses we provided are marked in red.

I’m pleased to continue review (round 2) the manuscript entitled “Genome-wide identification of the sweet potato [Ipomoea batatas (L.) Lam] DUF668 gene family and expression analysis under drought and salt stress” and recommend publishing it as a journal article.

Foremost the authors have improved the manuscript, in comparison with the last version, adding the necessary data as tables and figures, and in my view, remain some minor revisions that I would recommend handling before publication.

  1. Lines 83-86: This paragraph should be in the conclusion and perspectives section.

Thank you very much for your suggestion. We have moved this section to the conclusion section and updated it in this upload.

  1. Line 130: “cis-acting” instead of “cis-acting”.

Thank you very much for your suggestion. We have changed "cis-acting" to "cis-acting" and updated it in this upload.

  1. Lines 137-142: Very complex paragraph that need to be rewritten.

Thank you very much for your suggestion. We have rewritten this paragraph, simplified it, and updated it in this upload.

  1. Line 174: “Chr05” not “Chr04”.

Thank you very much for the suggestion. We have changed "Chr04" to "Chr05" and updated it in this upload.

  1. Line 175: “Chr15 contained 2 DUF668 genes” not 3.

Thank you very much for the suggestion. We have changed "3" to "2" and updated it in this upload.

  1. Line 216: “The majority of group 1 genes have 1 exon” not all of them.

Thank you very much for your suggestion. This sentence is indeed a mistake in our formulation, and we have made changes and updated it in this upload.

  1. Figure 2B: Not clear and should be improved.

Thank you very much for your suggestion. We have improved the clarity of Figure 2B and updated it in this upload.